# The Extent of Lifestyle-Induced Weight Loss Determines the Risk of Prediabetes and Metabolic Syndrome Recurrence during a 5-Year Follow-Up

**DOI:** 10.3390/nu14153060

**Published:** 2022-07-26

**Authors:** Silke Zimmermann, Mandy Vogel, Akash Mathew, Thomas Ebert, Rajiv Rana, Shihai Jiang, Berend Isermann, Ronald Biemann

**Affiliations:** 1Institute of Laboratory Medicine, Clinical Chemistry and Molecular Diagnostics, University of Leipzig, 04103 Leipzig, Germany; akash.mathew@medizin.uni-leipzig.de (A.M.); rajiv.rana@medizin.uni-leipzig.de (R.R.); shihai.jiang@medizin.uni-leipzig.de (S.J.); berend.isermann@medizin.uni-leipzig.de (B.I.); 2Centre for Pediatric Research, Department Woman and Child Health, Hospital for Children and Adolescents, University of Leipzig, 04103 Leipzig, Germany; mandy.vogel@medizin.uni-leipzig.de; 3Department of Clinical Science, Intervention and Technology, Division of Renal Medicine, Karolinska Institutet, SE-141 86 Stockholm, Sweden; thomas.ebert@medizin.uni-leipzig.de; 4Medical Department III—Endocrinology, Nephrology, Rheumatology, University of Leipzig Medical Center, 04103 Leipzig, Germany

**Keywords:** lifestyle-induced weight loss, metabolic syndrome, long-term benefit, 5-year follow-up, prediabetes

## Abstract

It is controversial whether lifestyle-induced weight loss (LIWL) intervention provides long-term benefit. Here, we investigated whether the degree of weight loss (WL) in a controlled LIWL intervention study determined the risk of prediabetes and recurrence of metabolic syndrome (MetS) during a 5-year follow-up. Following LIWL, 58 male participants (age 45–55 years) were divided into four quartiles based on initial WL: Q1 (WL 0–8.1%, *n* = 15), Q2 (WL 8.1–12.8%, *n* = 14), Q3 (WL 12.8–16.0%, *n* = 14), and Q4 (WL 16.0–27.5%, *n* = 15). We analyzed changes in BMI, HDL cholesterol, triglycerides (TGs), blood pressure, and fasting plasma glucose (FPG) at annual follow-up visits. With a weight gain after LIWL between 1.2 (Q2) and 2.5 kg/year (Q4), the reduction in BMI was maintained for 4 (Q2, *p* = 0.03) or 5 (Q3, *p* = 0.03; Q4, *p* < 0.01) years, respectively, and an increase in FPG levels above baseline values was prevented in Q2–Q4. Accordingly, there was no increase in prediabetes incidence after LIWL in participants in Q2 (up to 2 years), Q3 and Q4 (up to 5 years). A sustained reduction in MetS was maintained in Q4 during the 5-year follow-up. The present data indicate that a greater initial LIWL reduces the risk of prediabetes and recurrence of MetS for up to 5 years.

## 1. Introduction

Lifestyle-induced weight loss (LIWL), based on behavior therapy, is an important and effective strategy to manage metabolic syndrome (MetS) and prevent disease progression [1,2,3]. However, maintaining long-term weight loss (WL) is challenging, especially in individuals with MetS, since LIWL interventions typically result in early WL followed by progressive weight gain [1,4,5]. Hence, WL typically cannot be maintained in the long term [6]. It has been shown that 5 years after completing a structured LIWL program, the average WL maintained was only a 3.2% reduction from the initial weight [7]. Despite successful initial WL achievement, the majority of individuals regain weight in the long term after completing the LIWL intervention [8,9,10].

Because we hypothesized that a higher initial weight loss would have a positive impact on long-term outcome, the aim of this study was to determine whether the extent of WL achieved in participants of a structured LIWL program may predict WL maintenance, improvement in parameters characteristic of MetS, and recurrence of MetS during 5 years of follow-up in individuals who were diagnosed with MetS before the LIWL program and thus at a high risk of developing diabetes mellitus and cardiovascular disease [11,12]. 

To address these questions, we determined the changes in BMI, HDL cholesterol, systolic blood pressure (RR_sys_), diastolic blood pressure (RR_dia_), triglycerides (TGs), and fasting plasma glucose (FPG) before and after LIWL in 58 male participants with MetS at annual follow-up visits over a period of 5 years in a prospective study.

## 2. Materials and Methods

### 2.1. Research Design and Study Population

The study is embedded in a prospective, two-armed, controlled, monocentric, randomized, 6-month intervention trial aiming to identify changes in clinical and laboratory parameters in individuals with MetS following LIWL [13,14]. The trial was registered on the German Clinical Trials Register (ICTRP Trial Number: U1111-1158-3672). The trial included nonsmoking, nondiabetic men aged between 45 and 55 years with MetS, as defined by the consensus definition in 2009 [15]. Three out of the following five criteria needed to be met: abdominal obesity (waist circumference > 102 cm or BMI > 30 kg/m^2^); fasting TG concentration ≥1.7 mmol/L (or pharmaceutical intervention); high-density lipoprotein (HDL) cholesterol <1.00 mmol/L; FPG ≥ 5.6 mmol/L (or pharmaceutical intervention); and blood pressure ≥130/85 mmHg or treatment for hypertension. Exclusion criteria were smoking, type 2 diabetes mellitus, surgical procedure for WL within the previous 6 months, severe renal dysfunction (creatinine concentration >2.0 mg/dL), active liver disease, obesity of known endocrine origin, or inability to walk at least 30 min per day. Participants were recruited by an advertisement in a regional newspaper. Out of 133 individuals screened according to the inclusion or exclusion criteria from May 2012 to August 2012, 74 individuals were selected for the trial. All participants underwent a structured education program about diet and the importance of physical activity. Individuals were randomly assigned to a 6-month telemonitored LIWL program or a control arm as described previously [13,14,16]. Participants in the treatment arm were advised to lower their calorie intake by 500 kcal/day and to adopt a low-carbohydrate diet with a preference for low-GI carbohydrates [17]. Furthermore, participants were advised to increase their usual daily physical activity but to keep their pulse below 120 beats/min. Beyond these instructions, no special diet or recommendations, e.g., for physical activities, were given. During the LIWL intervention, participants recorded their daily body weight and received weekly written feedback commenting on their individual weight progress. Thirty participants in the control arm and thirty-three participants in the treatment arm completed the study [13]. During the follow-up, participants initially assigned to the control arm were offered enrollment in the treatment arm. Twenty-five individuals completed the subsequent 6-month LIWL program and received exactly the same intervention as participants in the initial treatment group. All 58 participants who completed the LIWL program were followed up for 5 years after WL. The study population did not differ in regard to the distribution of age, sex, or parameters of MetS.

### 2.2. Clinical and Laboratory Parameters

All blood samples were collected in the morning (8 a.m. to 9 a.m.) from the antecubital vein after a 12 h overnight fast. All laboratory measurements were performed at the Institute of Clinical Chemistry and Pathobiochemistry, OvGU, Magdeburg, Germany. FPG, TGs, and HDL cholesterol were analyzed by commercial enzymatic methods using a random access analyzer (Modular, Roche Diagnostics, Mannheim, Germany). FPG was determined in sodium fluoride plasma. HbA1c was determined by high-performance liquid chromatography.

### 2.3. Statistical Analysis

Data are given as the median and interquartile range (IQR). We have applied the D’Agostino-Pearson omnibus normality test. Since not all data were not normally distributed, nonparametric tests were applied for all statistical analyses. Differences between independent samples were analyzed by the Mann-Whitney U test. Paired samples were analyzed by the Wilcoxon signed-rank test. Categorical variables are given as frequencies, and McNemar’s test was used to compare differences between quartiles. Fisher’s exact probability test was used to analyze differences in group sizes. Trends in weight gain and in the related parameters were estimated using linear hierarchical regression models. To account for multiple measurements per subject, the subject was added as a random effect to the models. The association between all outcomes and time after LIWL was modeled as a quadratic after removing the cubic term because it was not significant. In addition, for mean yearly rates, the same associations were modeled linearly. The results were considered significant at *p* ≤ 0.05. The data analysis was performed using GraphPad Prism version 8.0 (GraphPad Software Inc., La Jolla, CA, USA) and R version 4.2 [18].

## 3. Results

### 3.1. Long-Term Effects of Lifestyle-Induced Weight Loss on BMI

Following LIWL, we annually analyzed parameters characteristic of MetS, such as HDL cholesterol, TGs, systolic (RR_sys_) and diastolic (RR_dia_) blood pressure, and FPG, during a 5-year follow-up period. In total, 58 nonsmoking men (45–55 years) with MetS participated in this long-term study (Figure 1).

Participants in the study were divided into four quartiles Q1–Q4 (Figure 1, Appendix A) based on WL after LIWL. During the 5-year follow-up period, several participants dropped out of the study for undisclosed reasons, resulting in an overall lower number of participants at years 4 and 5 (Table 1). At the last follow-up visit, 47%, 60%, 73%, and 87% of participants from Q1, Q2, Q3, and Q4 participated, respectively (Table 1), indicating that the participants’ motivation to attend the follow-up appointments was associated with the level of initial WL.

The median WL was 12.7% after completing the LIWL program and 3.7% at 5 years of follow-up. The median BMI reduction after LIWL in each quartile was 5.7% in Q1 (*p* < 0.01), 10.9% in Q2 (*p* < 0.01), 13.6% in Q3 (*p* < 0.01), and 19.1% in Q4 (*p* < 0.01) (Figure 2, Table 2). 

Weight gain after LIWL was 1.7, 1.2, 2.2, or 2.5 kg per year in Q1, Q2, Q3, and Q4, respectively, and differed significantly between Q1 and Q4 (*p* < 0.01) and between Q2 and Q4 (*p* = 0.05) over the whole follow-up period (Figure 3, Appendix A). 

While the BMI in participants from Q1 returned to baseline after 1 year, the reduction in BMI was maintained in participants from Q2 (up to 4 years) and in participants from Q3 and Q4, even until 5 years of follow-up (Figure 3, Table 2).

To analyze the relationship of initial WL with parameters characteristic of MetS, we next compared levels of HDL cholesterol, TGs, RR_sys_, and FPG between the WL quartiles Q1–Q4.

### 3.2. Long-Term Effects of Lifestyle-Induced Weight Loss on TGs

Following LIWL, a median TG reduction of 36.5% in Q2 (*p* < 0.01), 32.3% in Q3 (*p* < 0.01), and 47.4% in Q4 (*p* < 0.01) was observed (Figure 4a, Table 3). 

Since no difference was observed in Q1, the results suggest that the reduction in TG levels was dependent on the extent of WL (Figure 3). A high response to the LIWL program (Q4) was associated with a sustained reduction in mean TG levels over 4 years of follow-up, and even moderate WL (Q2) reduced mean TG levels up to 2 years after LIWL. The annual increase in TG levels after LIWL was comparable between Q2 and Q4, at 0.2 mmol/L per year (Appendix A). In contrast, no changes following LIWL or during the 5-year follow-up were observed in Q1 participants, suggesting that WL below 8.1% had no sustained impact on TG levels. Overall, the effect of LIWL on TG levels did not last as long as the sustained reduction in BMI. Regarding the intake of antilipidemic drugs, no changes were observed following LIWL or during the 5 years of follow-up (Appendix A).

### 3.3. Long-Term Effects of Lifestyle-Induced Weight Loss on HDL Cholesterol

In addition to reduced TG levels, the extent of LIWL was associated with increased levels of HDL cholesterol in Q2 (18.6%, *p* < 0.01), Q3 (21.3%, *p* < 0.01), and Q4 (17.9%, *p* < 0.01). In participants who lost less than 8.1% of their initial weight (Q1), HDL cholesterol remained unchanged following LIWL and even declined below baseline values after the third year of follow-up. Using linear approximation analysis, the decrease in HDL cholesterol after LIWL was 0.06, 0.04, 0.06, and 0.03 mmol/L per year in Q1 (*p* < 0.01), Q2 (*p* < 0.01), Q3 (*p* < 0.01), and Q4 (*p* < 0.01), respectively, with significant differences between Q3 and Q4 (*p* = 0.02) (Figure 4b, Appendix A). Hence, the increase in HDL cholesterol after LIWL was abolished immediately post LIWL (Q3) or after the first year of follow-up in participants in Q3 or Q4, respectively (Figure 4b, Table 4). However, HDL levels did not fall below baseline during the 5-year follow-up in Q2–Q4. Our results indicate that in individuals with MetS, a WL of more than 8.1% following a structured LIWL program was tentatively associated with higher HDL cholesterol over a 5-year follow-up period.

### 3.4. Long-Term Effects of Lifestyle-Induced Weight Loss on Systolic Blood Pressure (RR_sys_)

In contrast to other parameters of MetS, RR_dia_, and RR_sys_ remained more or less stable throughout the 5-year follow-up period. Some changes in RR_sys_ were observed, but these were inconsistent among the four quartiles overall. Thus, RR_sys_ decreased by 7.7% in Q2 (*p* < 0.01) and Q4 (*p* = 0.02, Figure 4c, Table 5), while no changes were observed in Q1 or Q3 immediately after LIWL.

The only quartile with persistently and partially significantly reduced RR_sys_ was Q4. Similar results were observed for RR_dia_ values. RR_dia_ remained largely stable during the follow-up period but were decreased in the 5-year follow-up in Q4 (*p* < 0.01, Appendix A). Analyses of blood pressure changes may have been confounded by the fact that nearly all participants were treated with blood pressure-lowering medication during the LIWL program and during the 5-year follow-up period (Appendix A). However, the number of participants treated with antihypertensive medication was lower at the 3- and 5-year follow-up in Q4 than in the other quartiles, indicating that high WL might be associated with beneficial long-term effects on blood pressure, as reflected by a reduced use of antihypertensive drugs. Hence, our data suggest that a pronounced reduction in BMI upon LIWL (>16.0%, as observed in Q4) results in a long-lasting reduction in RR_sys_ levels.

### 3.5. Long-Term Effects of Lifestyle-Induced Weight Loss on Fasting Plasma Glucose

An effect of LIWL on reduced FPG was only apparent in participants in Q2 and Q4, of 7.5% (*p* = 0.02) and 6.9% (*p* = 0.02), respectively, and remained significant only in Q4 for a follow-up period of 3 years (Figure 4d, Table 6).

Using linear approximation analysis, the increase in FPG after LIWL was 0.28 mmol/L per year in Q1 (*p* < 0.01), 0.16 mmol/L per year in Q2 (*p* < 0.01), 0.14 mmol/L per year in Q3 (*p* < 0.01), and 0.11 mmol/L per year in Q4 (*p* < 0.01, Appendix A, Figure 4d). Accordingly, FPG levels tended to increase over time and were significantly increased in participants in Q1 after 5 years of follow-up as compared to Q4 (*p* = 0.05). Importantly, an increase in FPG above baseline values was not pronounced in Q2–Q4 during the 5-year follow-up. Our results indicate that a WL of more than 8.1% following LIWL efficiently prevents an increase in FPG levels over a 5-year follow-up period.

### 3.6. Frequency of Prediabetes after Lifestyle-Induced Weight Loss

The diagnosis of prediabetes is based on glycated hemoglobin levels between 5.7 and 6.4% and FPG between 5.6 and 6.9 mmol/L [19]. The frequency of prediabetes before LIWL was comparable between Q1 and Q4, with 47% in Q1, 50% in Q2, 36% in Q3, and 47% in Q4. Following LIWL, the frequency of prediabetes was reduced to comparable levels in all quartiles (13% in Q1, 14% in Q2–Q3, and 7% in Q4), reflecting an immediate benefit across all quartiles. A sustained increase in prediabetes frequency after LIWL was apparent only in Q1 (*p* = 0.03, Table 7), which agreed with the increased FPG levels in Q1. In Q2, the frequency of prediabetes increased—with larger variations overall—while there was no increase in prediabetes incidence after LIWL in Q3 or Q4 during the 5-year follow-up, indicating that greater initial LIWL reduces the risk of prediabetes. During follow up, two participants started on metformin in year 4 and year 5 (Q1 and Q3, respectively). No other antidiabetic drugs were prescribed during the study period.

### 3.7. Long-Term Effect of Lifestyle-Induced Weight Loss on Metabolic Syndrome

An inclusion criterion for entering the study was the presence of MetS (prevalence 100% before LIWL). Following LIWL, the frequency of MetS was reduced to 73% in Q1 (*p* = 0.05), 43% in Q2 (*p* < 0.01), 29% in Q3 (*p* < 0.01), and 33% in Q4 (*p* < 0.01, Table 8). Based on the criteria for MetS, we determined the long-term effect of LIWL on MetS among the quartiles during the 5-year follow-up period. Although the long-term reduction in MetS frequency was maintained in Q4, the frequency of participants without MetS declined in Q1, Q2, and Q3 at 1 (Q1) or 2 (Q2, Q3) years after LIWL, respectively. Congruent with the increased frequency of prediabetes during the 5-year follow-up, the ability of LIWL to reduce the recurrence of MetS was weakest in Q1. Our results indicate that the improvement in MetS following LIWL strongly depends on the extent of initial WL, with long-lasting effects in individuals who lose more than 16% of their body weight (Q4).

## 4. Discussion

LIWL is regarded as an efficient method to reverse or ameliorate the metabolic changes associated with MetS [1,2,20]. However, WL is difficult to maintain over the long term [4]. The aim of this study was to determine (i) whether the initial amount of WL following LIWL is associated with long-term WL maintenance, (ii) the relationship between initial WL and the parameters characteristic of MetS, and (iii) whether the extent of initial WL in LIWL predicts the recurrence of prediabetes and MetS during 5 years of follow-up.

To that end, participants in a LIWL trial were divided based on the initial WL into four quartiles, Q1–Q4, with Q1 reflecting the lowest WL and Q4 reflecting the highest. The differentiation between the quartiles shows that participants with a higher initial WL (>Q1) were able to maintain their body weight below their baseline weight for up to 4 years (Q2) or 5 years (Q3, Q4) of follow up. Linear approximation revealed that annual weight regain was between 1.2 (Q2) and 2.5 (Q4) kg/year. Hence, our results indicate that the regain of body weight after LIWL occurs over time, which is congruent with other studies [6]. Despite successfully achieving WL, the majority of individuals regain the weight that was initially lost over the long term, when former food consumption habits and former lifestyle habits are resumed [8,9]. However, the linear weight gain in all quartiles in the current study suggests that a larger initial WL translates into a better outcome during follow-up and that such benefits are evident for at least 5 years. This is in line with a meta-analysis that examined the long-term WL maintenance in individuals completing a structured LIWL program [7]. According to the definition of successful WL maintenance of Wing and Hill [21], which was “intentionally losing at least 10% of initial weight and keeping it off for at least 1 year”, we found long-lasting effects in Q2–Q4 participants who lost more than 8.1% of their body weight.

In parallel with weight gain, there was a deterioration in characteristic parameters of MetS during the 5-year follow-up period. Although TG levels remained lower in participants in Q2 and Q3 for 1 year and in participants in Q4 up to 4 years following LIWL, the effect of reducing TG levels was absent in Q1, indicating that the degree of WL following LIWL determines the long-term increase in TG levels. Congruently, a reduction in FPG was only apparent in participants with the largest WL (Q4), and the FPG increase during the 5-year follow-up was most pronounced in participants with lower WL. It has been shown that increased TG levels may have detrimental effects on FPG levels [22]. One possible explanation for the FPG differences between Q1 and Q4 is that insulin sensitivity was impaired due to elevated TG levels [23,24,25,26,27,28]. Although participants in Q2–Q4 maintained their FPG levels close to baseline values after 5 years, an FPG increase above baseline values was apparent in participants in Q1. Accordingly, there was an increase in prediabetes incidence after LIWL during the 5-year follow-up in Q1. Since adipose tissue itself acts as an endocrine organ and produces numerous adipokines that may modulate glucose metabolism and insulin sensitivity, differences in body fat percentage may have influenced observed changes in FPG between Q1 and Q4 [29,30]. Overall, the amount of WL following LIWL and the ability to maintain WL are good predictors for the reversal of abnormalities in lipid and glucose metabolism in patients with MetS. Taken together, the current results indicate that WL should be approximately 8% or higher to achieve beneficial long-term effects after LIWL therapy with regard to body weight and TG and blood glucose levels.

Interestingly, the lowest long-term effects were observed in regard to serum HDL cholesterol. Low HDL cholesterol is a component of MetS and reflects an increased risk of cardiovascular disease progression. Despite successful WL maintenance, the beneficial HDL cholesterol increase after LIWL in Q2–Q4 was lost immediately or after 1 year of follow-up. The inverse association between HDL cholesterol and incident MetS is independent of traditional and MetS risk factors, obesity, and markers of insulin resistance [31], suggesting that low HDL cholesterol levels may be less representative of MetS and the associated cardiovascular risk. Indeed, studies show either a reduction or increase in HDL cholesterol after WL, indicating that the cardiovascular risk reduction after WL may be independent of HDL cholesterol levels [32]. Furthermore, genetic variants associated with reduced HDL cholesterol do not necessarily increase the risk of CVD [33]. Hence, plasma HDL cholesterol levels are apparently a poor surrogate marker for beneficial effects following LIWL [34].

A recent meta-analysis of randomized controlled trials showed that the effect of weight reduction on blood pressure depends on the amount of WL [35]. Unexpectedly, the RR_sys_ did not show quartile-dependent changes during follow-up in our study. One possible explanation might be the frequent use of antihypertensive drugs during the 5-year follow-up among the included study participants. However, the reduced number of participants treated with antihypertensive medication in Q4 together with lower RR_sys_ and lower RR_dia_ levels at the fifth year of follow-up suggests that pronounced reduction in the BMI upon LIWL might be associated with beneficial long-term effects on blood pressure.

Our results indicate that the amount of WL after LIWL is associated with long-term improvement in parameters characteristic of MetS, with the most pronounced effects in participants with an initial WL of more than 16% (Q4). Hence, the present data suggest that pronounced WL after LIWL may promote the transition of “metabolically unhealthy obesity”, which is associated with a higher risk of type 2 diabetes and cardiovascular disease, to “metabolically healthy obesity”, which is characterized by the absence of cardiometabolic abnormalities despite the presence of obesity [36]. The degree of WL following LIWL might be a good predictor for the metabolically healthy obesity transition. However, even when a metabolically healthy obese phenotype can be maintained over a long period of time, obesity remains a risk factor for cardiovascular disease [37]. It is a well-known phenomenon that responses to treatment are heterogeneous, which, in line with the existence of diverse obesities [38], may influence long-term outcomes following weight loss [39]. Our data are in agreement with the Look AHEAD study that examined the effects of lifestyle intervention in a large cohort of participants with type 2 diabetes. Here, the weight loss achieved during the first 2 months predicted long-term outcomes for up to 8 years [40,41]. However, our study was not designed to determine the weight loss already after 2 months and hence we did not determine the earliest timepoint that was predictive of long-term outcomes. Indeed, determining the earliest timepoint may aid implementation of therapies, as early adjustments are possible. The Look AHEAD and the current study both support the concept that optimizing the outcome (here, weight loss) within the first months translates into beneficial long-term effects [42,43].

An important cornerstone of the successful maintenance of weight loss is motivation. We observed an overall dropout rate of 30% during the 5-year follow-up period, which is in line with comparable long-term studies [7]. Of note, the participants’ willingness to participate in the follow-up appointments decreased, especially in the lower WL groups (Q1, Q2), indicating that participants with a high level of WL are more motivated and hence attend follow-up appointments. Assuming that participants with lower WL were preferentially lost to follow-up, we suspect that this may result in a selection bias that reduces the observed differences among quartiles.

The strengths of the current study include a well-characterized and well-matched study population at baseline, a prospective study design, and high compliance of participants during the LIWL program as a result of daily telemonitoring and weekly letters commenting on individual weight progress. All laboratory measurements were performed according to standard operating protocols, and laboratory technicians were blinded to the status of the samples. To generate a homogenous study group, only middle-aged male Caucasians with MetS were included in this study. Although homogeneity was increased among participants, the results cannot be generalized to other ethnic groups, sexes or individuals without MetS. Another limitation is the relatively small sample size and the drop-out of participants, especially in the lower WL quartile (Q1), over the 5-year follow-up period.

## 5. Conclusions

In conclusion, the current results indicate that the extent of weight reduction after LIWL determines the frequency of prediabetes development and the risk of recurrence of MetS during 5 years of follow-up.

## Figures and Tables

**Figure 1 nutrients-14-03060-f001:**
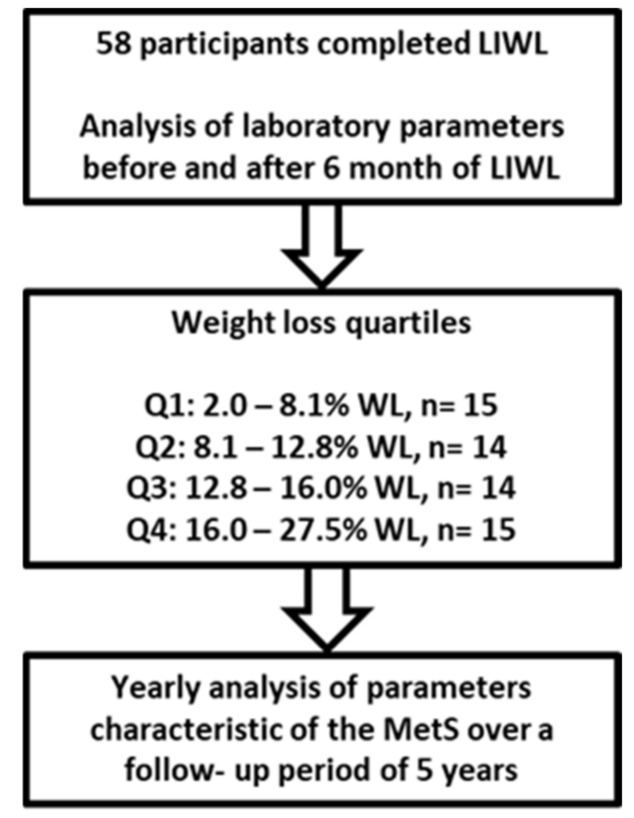
Schematic of study design. The study was embedded in a controlled, monocentric, 6-month lifestyle-induced weight loss (LIWL) trial. Paired blood samples were collected before and after LIWL as well as after each year of follow-up (total: 5 years of follow-up). For statistical analysis, participants were divided into four quartiles based on the extent of weight loss. Fifty-eight participants completed the LIWL program and were included in the data analysis.

**Figure 2 nutrients-14-03060-f002:**
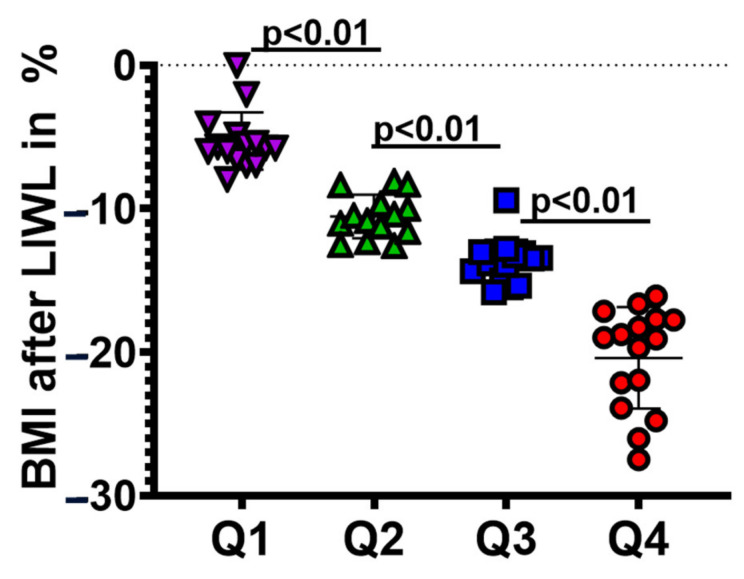
Division of study participants into four quartiles based on initial weight loss. Q1 represents the 25% of individuals with the lowest levels of weight loss. Differences between groups were analyzed by the Mann-Whitney U test.

**Figure 3 nutrients-14-03060-f003:**
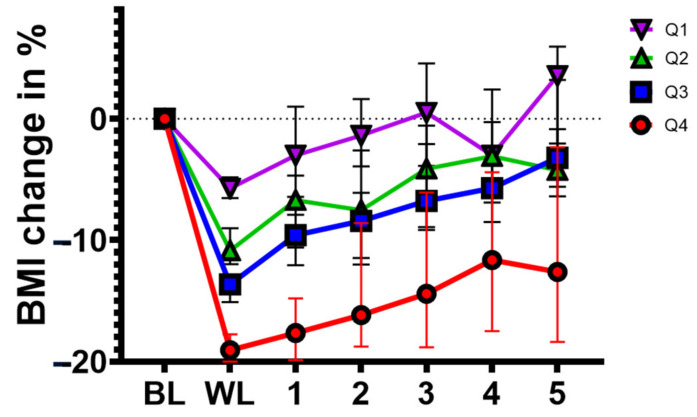
Absolute weight recovery is dependent on initial weight loss. Line plot summarizing changes in BMI during the 5-year follow-up in weight loss quartiles (Q1–Q4). Data are presented as the median (interquartile range). BL = baseline; WL = weight loss; 1–5 = years of follow-up.

**Figure 4 nutrients-14-03060-f004:**
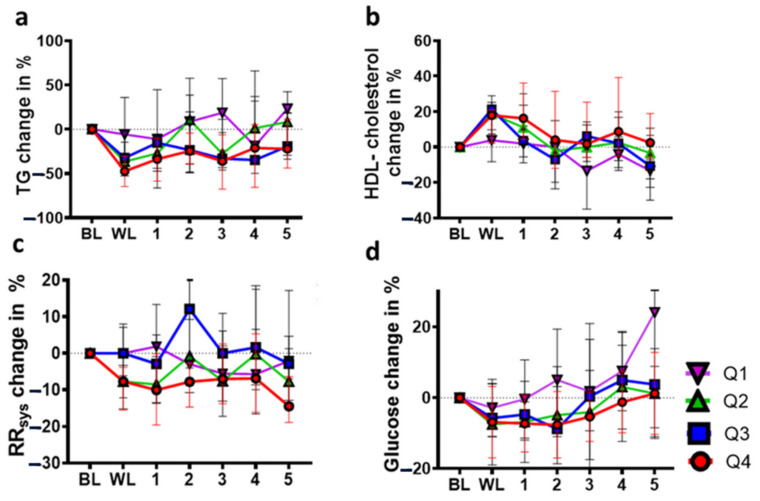
Long-term effects of lifestyle-induced weight loss on parameters of metabolic syndrome in weight loss quartiles (Q1–Q4). Triglyceride (TG) (**a**), HDL cholesterol (**b**), systolic blood pressure (RR_sys_) (**c**), and fasting plasma glucose (FPG) levels (**d**) during the 5-year follow-up. Data are presented as the median (interquartile range). BL = baseline; WL = weight loss; 1–5 = years of follow-up.

**Table 1 nutrients-14-03060-t001:** Table showing the absolute (*N*) and relative (%) numbers of participants in each quartile (Q1–Q4) at each timepoint.

	Q1 Total (*N*)/Relative (%)	Q2 Total (*N*)/Relative (%)	Q3 Total (*N*)/Relative (%)	Q4 Total (*N*)/Relative (%)	*p*
BL	15/100	14/100	14/100	15/100	
LIWL	15/100	14/100	14/100	15/100	
FU1	11/73	13/80	12/73	15/93	n.s.
FU2	8/53	12/73	12/73	13/80	n.s.
FU3	8/53	9/53	11/67	12/73	n.s.
FU4	6/40	11/67	12/73	13/80	0.02
FU5	7/47	10/60	12/73	14/87	0.01

Fisher’s exact probability test was used to analyze differences in group sizes (only significant values are listed). BL = baseline; LIWL = lifestyle-induced weight loss; FU1–5 = years of follow-up.

**Table 2 nutrients-14-03060-t002:** Median and interquartile ranges of BMI before and after LIWL and during the 5-year follow-up.

BMI (kg/m^2^)	BL	Q1	*p*	BL	Q2	*p*	BL	Q3	*p*	BL	Q4	*p*
LIWL	32.9 ± 6.6	30.9 ± 7.3	<0.01	33.0 ± 5.7	28.8 ± 4.9	<0.01	35.4 ± 3.4	30.0 ± 3.5	<0.01	35.2 ± 3.2	27.5 ± 5.1	<0.01
FU1		32.6 ± 6.2	n.s.		30.7 ± 7.2	<0.01		29.8 ± 3.2	<0.01		28.2 ± 5.1	<0.01
FU2		32.2 ± 4.1	n.s.		30.4 ± 6.7	<0.01		31.3 ± 4.4	<0.01		29.8 ± 6.4	<0.01
FU3		31.9 ± 5.4	n.s.		30.1 ± 4.1	n.s.		31.2 ± 3.7	<0.01		30.2 ± 7.8	<0.01
FU4		32.3 ± 5.0	n.s.		32.1 ± 4.7	0.03		32.4 ± 3.6	<0.01		31.2 ± 8.7	<0.01
FU5		33.8 ± 3.7	n.s.		33.6 ± 3.9	n.s.		33.6 ± 4.0	0.03		31.1 ± 8.7	<0.01

The Wilcoxon signed rank test was used to analyze changes between paired samples between BL and subsequent follow-up timepoints (only significant values are listed). BL = baseline; BMI = body mass index; LIWL = lifestyle-induced weight loss; FU1–5 = years of follow-up.

**Table 3 nutrients-14-03060-t003:** Median and interquartile ranges of TG levels before and after LIWL and during the 5-year follow-up.

TG (mmol/L)	BL	Q1	*p*	BL	Q2	*p*	BL	Q3	*p*	BL	Q4	*p*
LIWL	2.19 ± 2.2	2.16 ± 2.8	n.s.	2.0 ± 2.6	1.36 ± 0.8	<0.01	2.03 ± 1.9	1.4 ± 0.8	<0.01	1.83 ± 1.5	0.98 ± 0.4	<0.01
FU1		1.93 ± 1.8	n.s.		1.42 ± 2.1	0.03		1.37 ± 1.4	<0.01		1.10 ± 1.5	<0.01
FU2		2.31 ± 1.0	n.s.		1.46 ± 2.3	n.s.		2.01 ± 1.7	n.s.		1.4 ± 1.3	0.01
FU3		2.76 ± 1.8	n.s.		1.5 ± 1.4	n.s.		1.97 ± 1.2	n.s.		1.23 ± 1.6	0.02
FU4		2.46 ± 1.3	n.s.		1.8 ± 3.2	n.s.		1.95 ± 1.7	n.s.		1.25 ± 1.1	0.04
FU5		2.9 ± 1.7	n.s.		1.87 ± 2.4	n.s.		2.18 ± 2.3	n.s.		1.55 ± 1.0	n.s.

The Wilcoxon signed rank test was used to analyze changes between paired samples between BL and subsequent follow-up timepoints (only significant values are listed). BL = baseline; LIWL = lifestyle-induced weight loss; triglycerides = TG; FU1–5 = years of follow-up.

**Table 4 nutrients-14-03060-t004:** Median and interquartile ranges of HDL cholesterol levels before and after LIWL and during the 5-year follow-up.

HDL (mmol/L)	BL	Q1	*p*	BL	Q2	*p*	BL	Q3	*p*	BL	Q4	*p*
LIWL	1.41 ± 0.7	1.53 ± 0.8	n.s.	1.2 ± 0.4	1.33 ± 0.4	<0.01	1.23 ± 0.4	1.45 ± 0.6	<0.01	1.21 ± 0.4	1.46 ± 0.4	<0.01
FU1		1.6 ± 0.4	n.s.		1.26 ± 0.5	0.03		1.19 ± 0.7	n.s.		1.45 ± 0.4	<0.01
FU2		1.34 ± 0.7	n.s.		1.28 ± 0.5	n.s.		1.13 ± 0.7	n.s.		1.28 ± 0.6	n.s.
FU3		1.23 ± 0.5	0.02		1.28 ± 0.5	n.s.		1.24 ± 0.5	n.s.		1.21 ± 0.5	n.s.
FU4		1.38 ± 0.5	n.s.		1.32 ± 0.7	n.s.		1.13 ± 0.7	n.s.		1.27 ± 0.6	n.s.
FU5		1.24 ± 0.7	0.02		1.16 ± 0.5	n.s.		1.25 ± 0.4	n.s.		1.25 ± 0.4	n.s.

The Wilcoxon signed rank test was used to analyze changes between paired samples between BL and subsequent follow-up timepoints (only significant values are listed). BL = baseline; HDL = HDL cholesterol; LIWL = lifestyle-induced weight loss; FU1–5 = years of follow-up.

**Table 5 nutrients-14-03060-t005:** Median and interquartile ranges of RR_sys_ levels before and after LIWL and during the 5-year follow-up.

RR_sys_ (mmHg)	BL	Q1	*p*	BL	Q2	*p*	BL	Q3	*p*	BL	Q4	*p*
LIWL	140 ± 12	138 ± 16.0	n.s.	138 ± 14.0	122 ± 13.0	<0.01	140 ± 28.5	130 ± 14.0	n.s.	149.0 ± 18.0	130 ± 18.0	0.02
FU1		140 ± 23.8	n.s.		123 ± 18.8	0.03		125 ± 28.0	n.s.		135 ± 22.5	0.02
FU2		148 ± 57.3	n.s.		138 ± 20.0	n.s.		145 ± 25.0	n.s.		131 ± 23.3	n.s.
FU3		137 ± 35.0	n.s.		131.0 ± 10.0	n.s.		138 ± 25.0	n.s.		135 ± 25.0	n.s.
FU4		128 ± 27.3	n.s.		144.5 ± 33.3	n.s.		141 ± 30.0	n.s.		129.5 ± 31.3	n.s.
FU5		137 ± 12.3	n.s.		134 ± 16.0	n.s.		130 ± 18.0	n.s.		123 ± 17.5	<0.01

The Wilcoxon signed rank test was used to analyze changes between paired samples between BL and subsequent follow-up timepoints (only significant values are listed). BL = baseline; LIWL = lifestyle-induced weight loss; FU1–5 = years of follow-up; RR_sys_ = systolic blood pressure.

**Table 6 nutrients-14-03060-t006:** Median and interquartile ranges of fasting plasma glucose levels before and after LIWL and during the 5-year follow-up.

Glucose (mmol/L)	BL	Q1	*p*	BL	Q2	*p*	BL	Q3	*p*	BL	Q4	*p*
LIWL	5.9 ± 0.8	5.8 ± 0.7	n.s.	6.2 ± 1.2	5.7 ± 0.6	0.02	5.9 ± 1.2	5.4 ± 0.6	n.s.	6.2 ± 0.9	5.6 ± 0.7	0.02
FU1		5.9 ± 0.2	n.s.		5.6 ± 0.9	n.s.		5.6 ± 0.7	n.s.		5.6 ± 0.5	0.02
FU2		6.3 ± 0.7	n.s.		5.7 ± 1.0	n.s.		5.4 ± 0.7	n.s.		5.4 ± 0.9	0.03
FU3		6.1 ± 1.8	n.s.		5.7 ± 1.1	n.s.		5.7 ± 1.4	n.s.		5.7 ± 0.9	0.02
FU4		6.4 ± 1.0	n.s.		6.4 ± 1.2	n.s.		6.3 ±1.5	n.s.		5.8 ± 1.3	n.s.
FU5		6.8 ± 1.5	0.05		6.2 ± 1.4	n.s.		6.0 ± 1.2	n.s.		5.9 ± 0.8	n.s.

The Wilcoxon signed rank test was used to analyze changes between paired samples between BL and subsequent follow-up timepoints (only significant values are listed). BL = baseline; LIWL = lifestyle-induced weight loss; FU1–5 = years of follow-up.

**Table 7 nutrients-14-03060-t007:** Table showing absolute numbers of participants with prediabetes (PD) after LIWL during 5 years of follow-up and the total number of participants at the indicated timepoints in each quartile (Q1–Q4).

	Q1 PD/Total	*p*	Q2 PD/Total	*p*	Q3 PD/Total	*p*	Q4 PD/Total	*p*
BL	7/15		7/14		5/14		7/15	
LIWL	2/15		2/14		2/14		1/15	
FU1	4/11	n.s.	3/12	n.s.	1/11	n.s.	1/14	n.s.
FU2	2/8	n.s.	3/11	n.s.	2/11	n.s.	1/12	n.s.
FU3	3/8	n.s.	4/8	0.05	3/10	n.s.	3/11	n.s.
FU4	2/6	0.03	4/10	n.s.	3/11	n.s.	1/12	n.s.
FU5	3/7	0.03	3/9	n.s.	3/11	n.s.	1/13	n.s.

McNemar’s test was used to analyze differences between frequencies at different timepoints in each quartile compared to the frequency of PD after LIWL (only significant values are listed). BL = baseline; LIWL = lifestyle-induced weight loss; FU1–5 = years of follow-up; alpha level of significance = 0.05.

**Table 8 nutrients-14-03060-t008:** Table showing the absolute numbers of participants with metabolic syndrome (MetS) and the total number of participants at the indicated timepoints in each quartile (Q1–Q4).

	Q1 MetS/Total	*p*	Q2 MetS/Total	*p*	Q3 MetS/Total	*p*	Q4 MetS/Total	*p*
BL	15/15		14/14		14/14		15/15	
LIWL	11/15	0.05	6/14	<0.01	4/14	<0.01	5/15	<0.01
FU1	6/11	0.03	5/12	<0.01	4/11	<0.01	4/14	<0.01
FU2	7/8	n.s.	7/11	0.05	7/11	0.05	7/12	0.03
FU3	6/8	n.s.	6/8	n.s.	8/10	n.s.	8/11	n.s.
FU4	5/6	n.s.	10/10	n.s.	9/11	n.s.	9/12	n.s.
FU5	6/7	n.s.	9/9	n.s.	8/11	n.s.	9/13	0.05

McNemar’s test was used to analyze differences between frequencies at different timepoints in each quartile compared to the frequency of MetS at BL (only significant values are listed). BL = baseline; LIWL = lifestyle-induced weight loss; FU1–5 = years of follow-up; alpha level of significance = 0.05.

## Data Availability

Not applicable.

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
