# Peer review of "The Extent of Lifestyle-Induced Weight Loss Determines the Risk of Prediabetes and Metabolic Syndrome Recurrence during a 5-Year Follow-Up"

_nutrients, 2022, doi:10.3390/nu14153060_

Round 1

Reviewer 1 Report

Dear authors thank you for the opportunity to review this very interesting study. The manuscript is well written and all the results are clearly presented and discussed. The main limitation is the small sample size.

Author Response

PLease see the attachement. 

Reviewer 2 Report

GENERAL COMMENTS

The topic is interesting as it provides important insight into lifestyle-induced weight loss and the impact of recurrence during a 5-year follow-up on the risk of prediabetes and metabolic syndrome. In this sense, the manuscript addresses a worthwhile topic which has clinical relevance. However, some relevant aspects would need to be improved. 

The manuscript may benefit from considering the following aspects:

It would be useful and more informative to indicate in the abstract the main statistical differences.

Introduction, page 1, line 38: replace “maintained for” by “maintained in”.

Introduction: formulate the working hypothesis – what did you expect and why.

Materials and Methods, page 2, line 57: refernces jump from #10,11 to #28,29; references 12-27 are missing. In addition, the indicated references do not match the content.

Materials and Methods, page 2, line 58: justify why only males were included in the study.

Statistical analysis: why were non-parametric tests applied? Data did not follow a normal distribution? Which normality test was used? This information needs to be indicated.

Results` section: include a Table with the main clinical characteristics of participants in each quartile at baseline and thereafter to have a global picture (weight, age, BMI, basal glucose, insulin concentrations, etc).

Results: indicate pharmacological treatments (antihypertensives, antidiabetics, antilipidemics, etc) that were taken by participants in each Q as well as how intake decreased with WL during treatment and follow-up.

Results, page 3, line 110: explain abbreviature of RRsys

Results, page 3, lines 118-123: redundant information – already shown in Fig 1.

Results, page 4, line 131-132: p<0.01 in text while p<0.001 according to Figure 2 legend. In addition, the text is redundant with the Figure.

Results, page 7: why were only systolic blood pressure values mentioned? What happened to diastolic blood pressure values?

Analyze the differential characteristics between participants that lost less weight vs those that exhibited the highest WL.

Discussion

In this context, the Look-AHEAD trial should be mentioned. The Look AHEAD study has successfully randomised a large cohort of participants with type 2 diabetes with a wide distribution of age, obesity, ethnicity and racial background that examined the effects of lifestyle intervention. One- and two-month weight loss (WL) was associated with 8-year WL (Unick JL, Neiberg RH, Hogan PE, Cheskin LJ, Dutton GR, Jeffery R, Nelson JA, Pi-Sunyer X, West DS, Wing RR; Look AHEAD Research Group. Weight change in the first 2 months of a lifestyle intervention predicts weight changes 8 years later. Obesity (Silver Spring). 2015 Jul;23(7):1353-6). Weight losses in lifestyle interventions are variable, yet prediction of long-term success is difficult. The utility of using various weight loss thresholds in the first 2 months of treatment for predicting 1-year outcomes was examined (Unick JL, Hogan PE, Neiberg RH, Cheskin LJ, Dutton GR, Evans-Hudnall G, Jeffery R, Kitabchi AE, Nelson JA, Pi-Sunyer FX, West DS, Wing RR; Look AHEAD Research Group. Evaluation of early weight loss thresholds for identifying nonresponders to an intensive lifestyle intervention. Obesity (Silver Spring). 2014 Jul;22(7):1608-16). Given the association between initial and 1-year WL, the first few months of treatment may be an opportune time to identify those who are unsuccessful Unick JL, Beavers D, Jakicic JM, Kitabchi AE, Knowler WC, Wadden TA, Wing RR; Look AHEAD Research Group. Effectiveness of lifestyle interventions for individuals with severe obesity and type 2 diabetes: results from the Look AHEAD trial. Diabetes Care. 2011 Oct;34(10):2152-7). Intensive lifestyle intervention can produce sustained WL and improvements in fitness, glycemic control, and CVD risk factors in individuals with type 2 diabetes (Look AHEAD Research Group, Wing RR. Long-term effects of a lifestyle intervention on weight and cardiovascular risk factors in individuals with type 2 diabetes mellitus: four-year results of the Look AHEAD trial. Arch Intern Med. 2010 Sep 27;170(17):1566-75).

Heterogeneity in response to treatment is a well-known phenomenon, which also needs to be acknowledged in line with the existence of diverse obesities (ref Yárnoz-Esquiroz P, Olazarán L, Aguas-Ayesa M, et al. 'Obesities': Position statement on a complex disease entity with multifaceted drivers. Eur J Clin Invest. 2022 Jul;52(7):e13811). Include some references for the statement made in the first sentence; e.g. Catalán V, Avilés-Olmos I, Rodríguez A, et al. Time to Consider the "Exposome Hypothesis" in the Development of the Obesity Pandemic. Nutrients. 2022 Apr 12;14(8):1597 // 

In general, many references seem outdated and it would be better to cite more recent papers.

Page 11, line 296: a further explanation for the differences in FPG between 1 and Q4 may also underlie differences in body fat percentage (ref Frühbeck G, Gómez-Ambrosi J. Control of body weight: a physiologic and transgenic perspective. Diabetologia. 2003 Feb;46(2):143-72). If body composition measurements were not performed, this potential explanation should be at least mentioned in the Discussion.

References

Consider indications given above.

Reviewer 3 Report

1. Introduction: The author should introduce the significance of the research in more detail.

2. Line 226-228, Fig 4d: the result of Q1 in fifth year should be further discussed.

3. In Tables: Significant figures should be unified.

4. WL in Q4: 16.0-27.5%. Is this range too large? Many results are involved in Q4 in this paper. Will the result be affected if the range is too large?

Round 2

Reviewer 2 Report

The authors have successfully addressed all the issues raised.